# Epigenetic Mechanisms in Fabry Disease: A Thematic Analysis Linking Differential Methylation Profiles and Genetic Modifiers to Disease Phenotype

**DOI:** 10.3390/cimb47100855

**Published:** 2025-10-16

**Authors:** Jatinder Singh, Paramala Santosh, Uma Ramaswami

**Affiliations:** 1Department of Child and Adolescent Psychiatry, Institute of Psychiatry, Psychology and Neuroscience, King’s College London, London SE5 8AF, UK; 2Centre for Interventional Paediatric Psychopharmacology and Rare Diseases (CIPPRD), South London and Maudsley NHS Foundation Trust, London SE5 8AZ, UK; 3Centre for Interventional Paediatric Psychopharmacology (CIPP) Rett Centre, Institute of Psychiatry, Psychology and Neuroscience, King’s College London, London SE5 8AF, UK; 4Lysosomal Storage Disorders Unit, Royal Free London NHS Foundation Trust, London NW3 2QG, UK; 5Genetics and Genomic Medicine, University College London, London WC1N 1EH, UK

**Keywords:** Fabry disease, methylation, genetic modifiers, epigenome, genotype–phenotype correlations

## Abstract

**Background****/Objectives**: Fabry disease is an X-linked lysosomal storage disorder. It is characterised by impaired metabolism of glycosphingolipids whose accumulation causes irreversible organ damage and life-threatening complications. Genotype–phenotype correlations have a limited scope in Fabry disease as the disorder presents with wide-ranging clinical variability. In other X-linked disorders, epigenetic profiling has identified methylation patterns and disease modifiers that may explain clinical heterogeneity. In this narrative review and thematic analysis, the role of DNA methylation and epigenetics on the clinical phenotype in Fabry disease was investigated. **Methods**: Embase, PubMed, and PsycINFO were searched to identify literature on DNA methylation and epigenetics in Fabry disease. Based on the eligibility criteria, 20 articles were identified, and a thematic analysis was performed on the extracted data to identify themes. **Results**: Three themes emerged: (I) genetic modifiers, (II) methylation profiling, and (III) insights into X chromosome inactivation (XCI). The evidence synthesis revealed that telomere length, especially in early disease stages, bidirectional promoter (BDP) methylation by sphingolipids, epigenetic reader proteins, mitochondrial DNA haplogroups, and DNA methylation of the promoter region of the *calcitonin receptor* gene are potential genetic modifiers in Fabry disease. Methylation patterns also reveal episignatures in Fabry disease evolution and genes implicated in the maintenance of basement membranes. Studies on XCI further emphasise disease heterogeneity and draw attention to methodological issues in the assessment of XCI. **Conclusions**: This thematic review shows that DNA methylation and genetic modifiers are key factors modifying clinical variability in Fabry disease. More broadly, it underscores a crucial role for epigenetic processes in driving disease onset, progression, and severity in X-linked disorders.

## 1. Introduction

The X chromosome has more than 1000 genes essential to biological functions [1]. To prevent the deleterious consequences of a double dose of X-linked genes, early during embryogenesis, the X chromosome is randomly inactivated in humans. This epigenetic process is known as X chromosome inactivation (XCI) [2,3]. Consequently, XCI results in a random ‘mosaic’ pattern of active and inactive X chromosomes in cells throughout the body [1]. The inactive X chromosome exhibits increased levels of DNA methylation and histone modifications [4]. During the process of XCI, several hundred genes are silenced [5]. The implications of XCI have long been recognised in neurological diseases [6], but it can also assist in the identification of X-linked disease carriers [7]. Skewing (or the XCI ratio) occurs when one X chromosome is preferentially inactivated over the other and can happen in neurotypical females [8]. Theoretically, XCI ratios can range from a 0:100 to a 50:50 ratio [9]. Objective evidence has indicated that only a small number of neurotypical females show skewed patterns [9]. The clinical significance of skewing in neurotypical females is unclear, and those with highly skewed patterns may warrant further attention [9]. The XCI pattern can influence disease manifestation, especially when the mosaic pattern is skewed. In X-linked disorders, a skewed XCI pattern has clinical importance and confers a selective advantage to cells, especially when the X chromosome containing the mutated gene is inactivated.

There are more than 141 disorders associated with intellectual disability due to pathogenic variants on the X chromosome [10]. One such disorder is Rett syndrome (RTT) (OMIM #312750), a debilitating X-linked neurodevelopmental disorder. In RTT, skewing was observed in 36% (*n* = 45/125) in individuals with classical RTT [11,12]. Because the pathogenic variant (methyl-CpG binding protein 2, *MECP2*) responsible for most cases of classical RTT is located on the X chromosome, it implies that XCI could theoretically regulate the clinical severity of RTT. However, the impact of XCI upon disease severity in RTT is unclear. Evidence suggests that XCI in RTT does not influence clinical severity [12,13], while others suggest a correlation [14]. More recent research suggests that XCI has a minor impact on disease severity in those with severe *MECP2* pathogenic variants [15]. In Fabry disease (OMIM #301500), the mutated gene, *alpha-galactosidase A* (*GLA*), is also found on the X chromosome (Xq21.3-q22) [16]. The *GLA* gene encodes a lysosomal alpha-galactosidase A (α-GAL A) enzyme that is involved in the metabolism of glycosphingolipids, usually globotriaosylceramide (Gb3) and its deacetylated form (Lyso-Gb3). Enzyme deficiency causes a build-up of these glycosphingolipids in lysosomes, leading to systemic disease. More than 1000 pathogenic variants have been identified in the *GLA* gene [17], and the clinical phenotype depends upon the disease trajectory, i.e., classic and later onset forms, the type of *GLA* variant, gender, age, and the amount of α-Gal A activity [18].

Although Fabry disease is a metabolic disorder and RTT is a neurodevelopmental disorder, both disorders share commonalities. First, the build-up of Gb3 and Lyso-Gb3 deposits activates pro-inflammatory pathways and induces oxidative stress in Fabry disease [19,20]. This inflammatory burden is also mirrored in RTT, where individuals exist in a subclinical chronic inflammatory state [21]. Second, males are more severely impacted in Fabry disease [16]. While RTT predominantly affects females, it has also been identified in males with variable clinical presentation ranging from mild to severe [22]. Third, both Fabry disease and RTT have multi-organ involvement and are susceptible to life-threatening neurological complications [22,23], and fourth, both are characterised by genotype–phenotype heterogeneity. Indeed, RTT remains a clinical diagnosis [24], and genetic modifiers in RTT can influence disease progression [25,26]. Dysregulated methylation is a hallmark feature of RTT [27,28,29].

Skewing may occur through various mechanisms [30], including changes in DNA methylation [31]. The methylation status of the androgen receptor is the gold standard method for determining the XCI pattern [32]. Ultra-deep methylation analyses could help to facilitate our understanding of disease predisposition in X-linked disorders, especially in females with Fabry disease [33], where disease onset and clinical phenotype are more variable [34,35]. Nevertheless, little emphasis has been put on how dysregulated methylation may contribute to disease phenotype in Fabry.

###  Aim of the Narrative Review 

Given the clinical heterogeneity of Fabry disease and its association with DNA methylation, it would be essential to examine how changes in the Fabry epigenome influence the clinical profile of individuals. This notion is also shared by others who have suggested that DNA methylation may play a role in influencing the variability of the clinical phenotype in hemizygous and heterozygous Fabry disease [16]. Furthermore, because the clinical phenotype in Fabry disease individuals cannot be explained entirely by skewed XCI patterns, it would be sensible to consider other elements that could have a role, such as genetic modifiers of disease severity, as noted for other X-linked disorders [25,36,37]. The overarching aim of the present review was, therefore, to answer the following research question: What is the role of DNA methylation and genetic modifiers on the clinical phenotype in Fabry disease? To answer this question, the study reviewed the literature on methylation studies in Fabry disease. It had the following objectives: (I)To evaluate studies of DNA methylation in Fabry disease.(II)To perform a thematic analysis and identify themes that lead to a better understanding of clinical severity and disease prediction.(III)To determine whether areas emerging from the thematic analysis can be applied to other X-linked disorders.

## 2. Methods

### 2.1. Narrative Review Strategy

The structure of this narrative review was guided by a 12-step framework described in Kable et al. (2012) [38]. This framework is a step-by-step guide to undertake reviews and was recently adopted in a narrative review of communication abilities in RTT [39]. Considering this, the 12-step framework was used to improve the methodological rigour of the current review. In line with others [39], the current narrative review also adopted the Scale for the Assessment of Narrative Review Articles (SANRA) [40]. The SANRA is a scale for the quality appraisal of narrative reviews. It has the following six items: (I) justification of the review’s importance, (II) aims of the review, (III) description of the search, (IV) supporting references, (V) scientific reasoning, and (VI) presentation of data. The SANRA can be used as a self-assessment measure to ensure the robustness and improve the quality of narrative reviews [40]. There are no consensus guidelines for conducting narrative reviews; however, the SANRA criteria have been employed previously to enhance the quality and reliability of a narrative review [39]. Similarly, to improve the robustness of the evidence synthesis, the method of this narrative review was assessed against the SANRA criteria. The SANRA evaluation of the study is presented in Appendix A.

While this current narrative review did not follow the guidelines adopted for comprehensive systematic reviews, it did utilise aspects from the Preferred Reporting Items for Systematic Reviews and Meta-Analyses (PRISMA) guidelines [41] to screen and identify eligible articles as previously described [42]. The Embase, PubMed, and PsycINFO databases were searched in August 2025 with the age date filter restricted to the last 10 years (2015). Boolean operators and the truncation symbol (*) were used in the search. The ‘snowballing’ approach, which involves examining the reference sections of articles, was also used to identify any other relevant articles.

### 2.2. Search Terms

The following search terms were used:

((Fabry disease) OR (Fabry’s disease) OR (alpha-galactosidase A deficiency*) OR (Anderson-Fabry disease)) AND ((methylation) OR (DNA) OR (methylome) OR (epigenome) OR (epigenetic*) OR (Histone modification*)).

### 2.3. Population Characteristics

The database searches focused exclusively on studies performed in individuals with Fabry disease, specifically on methylation or aspects related to methylation, such as histone modification.

### 2.4. Eligibility Criteria

Inclusion criteria are as follows:➢Peer-reviewed full-text articles in the English language.➢Participants with a diagnosis of Fabry disease. This can also include studies performed on cell lines derived from individuals with Fabry.➢The study included articles from 2015 to August 2025.

Exclusion criteria are as follows:➢Studies performed exclusively using animal models.➢Book chapters, conference abstracts, meta-analyses, preprints, protocols, and all types of review articles.

### 2.5. Extraction of Data

The following data were extracted from the eligible articles and documented in a table: sample size, study purpose, participant characteristics, methods used for assessing methylation, and summary of the key findings. Data extraction was performed by the lead author (J.S.). When the data had been extracted into a summary table, it was reviewed by the third author (U.R.). Any issues were resolved before the table of extracted data was finalised.

### 2.6. Thematic Analysis Procedure

Following data extraction, a thematic analysis was performed by the first author (J.S.). The thematic analysis framework was reviewed by another author (U.R.), and a consensus was reached before the framework was approved. The thematic analysis approach used the exemplar described in Naeem et al. (2023) [43], which followed a structured approach: (I) select statements, (II) choose keywords, (III) perform coding, and (IV) generate themes.

(I)Selection of Statements

In this process, a grounded-theory approach was used as a recognised method for identifying themes directly from data [44]. Statements were extracted from the 20 studies with the aim of identifying keywords.

(II)Keywords

This process enhances the methodical rigour of the thematic analysis, as the selection of keywords from statements improves the conceptualisation stage of the analysis [43]. Colour was used in the statements to select keywords that identified essential concepts from the studies, guided by the 6Rs selection of keywords of thematic analysis in complex datasets [43].

(III)Inductive Coding

Coding was performed manually by J.S. to capture the essential attributes of the data relevant to the research question ‘*What is the role of DNA methylation and genetic modifiers on the clinical phenotype in Fabry Disease*?’ By examining the interconnection and patterns between the keywords, succinct words/terms (codes) were formulated and categorised to identify patterns in the data. This process of inductive coding is data-driven, allowing themes to emerge from the data [45].

(IV)Generation of Themes

Codes were grouped into meaningful themes that linked the research question to the data. The 4Rs framework, consisting of ‘reciprocal, ‘recognisable’, ‘responsive’, and ‘resourceful’, as described in Naeem et al. (2023) [43], was used to assist in the identification of themes. The frequency of each theme was summarised in Microsoft Excel.

The thematic analysis framework used in this study to identify the themes is described in Appendix A.

## 3. Results

### 3.1. Study Characteristics

To cover as much of the literature as possible related to methylation, the words methylome, epigenome, epigenetic, and histone modification were added to make the search as expansive as possible. Following a search of three databases (Embase, PubMed, and PsycINFO), the titles and abstracts of 567 articles were screened. When assessed for eligibility, 549 records were excluded (425 from Embase, 121 from PubMed, and 3 from PsychINFO), and 18 articles met the eligibility criteria. An additional two articles were included following a ‘snowball’ search, and in total, 20 articles were included and analysed (Figure 1 and Table 1). Three studies were case reports that reported on Fabry patients with unique clinical presentations [46,47,48]. A severe phenotype was observed in a female with associated 10q26 deletion syndrome [46]. Males usually have a more severe presentation; however, a late-onset male patient with residual α-galactosidase activity but milder organ involvement was reported [47]. A further case identified a new pathogenic variant (c.270C>G [*p.Cys90Trp*]) in Fabry disease [48]. These cases highlight the clinical complexity of Fabry disease in hemizygous and heterozygous individuals. The sample size of other studies ranged from 4 [49] to 510 individuals [50]. Two studies used fibroblast or endothelial cell lines derived from individuals with Fabry disease to investigate autophagy [51] and DNA methylation patterns [52]. A range of assessment methods was used for evaluating XCI and DNA methylation patterns. These included studies that used the Human Androgen Receptor (HUMARA) test [53,54,55] to bisulfite treatment [46,52,56]. The FAbry STabilisation indEX (FASTEX) is a tool used to assess clinical severity in Fabry disease [57]. Two studies have assessed clinical severity using FASTEX [54,58]. Another used the Mainz Severity Score Index (MSSI) and the Fabry disease severity scoring system to assess changes in clinical severity [59].

The studies exhibited heterogeneity across multiple domains. The design of the studies varied from case reports to cross-sectional and observational studies. While most studies included human subjects, some also utilised cell lines derived from individuals with Fabry disease [51,52]. Studies also demonstrated methodological variation when assessing clinical severity. The FASTEX score was used in two studies [54,58], while another used the MSSI and the Fabry disease severity scoring system [59]. Different research methods were used, including haplogroup typing of mitochondrial DNA [60] or sequencing and enzyme assays for phenotype classification [61]. Outcome heterogeneity was also evidenced from studies examining methylation status [62], telomere length [63], inflammatory gene expression [64], and XCI patterns [65,66].

**Table 1 cimb-47-00855-t001:** Study characteristics.

Source	*N* (Fabry)	Study Purpose	Sample Characteristics	Relevant Findings
[46] Hossain et al. (2017)	1	Reports on a severe and unique clinical presentation of a female patient with Fabry disease	37-year-old female Patient had a family history of Fabry disease	The case report demonstrated a chromosome 10q26 deletion associated with Fabry disease. Methylation of the CpG island in the *GLA* gene was reported. The study suggested that methylation could be associated with early onset and disease severity in patients with Fabry.
[47] Bae et al. (2020)	1	Case of a male Fabry patient with a de novo somatic mosaicism with mild symptoms but classic *GLA* variant	The male patient first presented at the age of 34 years old and then revisited the hospital for ERF when 51 years old No family members had Fabry disease symptoms	The case report identified a p.Gly274Arg mutation. The mutant allele was found in 58% of buccal cells, 84% in blood, and 85% in urine. This case showed that Fabry disease was caused by a de novo somatic mutation of the *GLA* gene and suggests that somatic mosaicism could be an important disease modifier in Fabry disease.
[48] Čerkauskaitė et al. (2019)	^¥^ 4	Identification of the novel *GLA* gene mutation in a female with Fabry disease	49-year-old female with Fabry disease (case presentation) A heterozygous mutation in exon 2 of GLA, *c.270C>G* (p. Cys90Trp) was identified	The study identified a novel mutation *c.270C>G* (p. Cys90Trp) in the GLA gene. This mutation was associated with cardiac involvement. As this is a novel variant, the symptoms arising from this variant in classical Fabry disease are unclear. Further research needs to be performed to better understand the clinical manifestations caused by this variant.
[49] Al-Obaide et al. (2022)	4	Investigating the cumulative effects of *GLA* mutation and BDP methylation on disease severity	Four patients with Fabry disease aged 18 to 39 years (two male and two female) Control groups had four healthy (two male and two female) individuals	The study showed the cumulative effects of *GLA* mutation and BDP methylation on disease severity in three patients. The authors indicate that BDP is an additional genetic factor modulating disease severity alongside *GLA* mutations in patients with Fabry disease. However, this finding needs to be replicated in studies with larger samples.
[50] Sezer & Ceylaner (2021)	510	Genetic management algorithm for high-risk patients with Fabry disease	Male (229/510); mean age was 40.8 ± 15.0 Female (281/510); mean age was 39.7 ± 15.5) Patients had a confirmed diagnosis of Fabry disease	The study used a genetic algorithm to improve the management of Fabry disease and emphasised the need for an early diagnosis. The genetic algorithm can be used to assess whether the variant in FD is pathogenic or likely pathogenic.
[51] Yanagisawa et al. (2019)	^Ψ^ N/A	To assess whether GLA expression levels were associated with autophagy	Fibroblasts were obtained from a female patient severely affected with Fabry disease and two siblings (sisters) who had mild symptoms	The study showed that autophagy was abnormal in the female patient with severe disease, unlike the two sisters who had few symptoms and normal autophagic flux. Lysosomes were enlarged in the patient with severe disease. Overall, the study showed an association between clinical severity, dysregulated autophagy, and methylation of wild alleles in Fabry disease.
[52] Shen et al. (2022)	N/A	Investigating whether dysregulated DNA methylation has a role in the development of Fabry disease	Endothelial cell line from a patient with Fabry disease (R112H mutation, aged 64 years)	The study demonstrated that 15 signaling pathways could be impacted by methylation. In 21 genes that had differential methylation profiles, *collagen type IV alpha 1* and *alpha 2* genes showed hypomethylation. Methionine levels were elevated in cells from the Fabry patient. The study also demonstrated that abnormalities in glycolipid metabolism affect DNA methylation in patients with Fabry disease.
[53] Iza et al. (2025)	7	To examine the relationship between methylation and clinical disease in females with Fabry disease	The age range of the females was 10 to 70 All females were heterozygous, were from two families, and had a diagnosis of Fabry disease Six cases had the HUMARA assay	Methylation assays can be helpful in providing insights into females with Fabry disease. The study showed that the *GLA* gene does not consistently escape XCI in peripheral blood. No correlation was found between XCI and clinical severity scores. The authors highlighted the importance of selecting the relevant tissues for XCI analysis.
[54] Rossanti et al. (2021)	9	Examining whether the existence of skewed XCI in females with heterozygous pathogenic variants in the *GLA* gene affects the phenotype	Nine female patients with Fabry disease Age range: 40 to 75 years Average age of onset: 37.5 years	Methods included detection of pathogenic *GLA* variants using NGS, RNA sequencing, HUMARA assay, and FASTEX clinical severity scores. Except for one patient, no XCI skewing was observed in the remaining patients. The main findings from the study demonstrated that XCI skewing cannot explain the clinical severity in Fabry disease.
[55] Juchniewicz et al. (2018)	12	Analyze XCI patterns and examine their role in disease manifestation in female patients with Fabry disease	Mean age (range) of 12 heterozygous females with a family history of Fabry disease: 40.7 years (11 to 67 years) Patient clinical phenotype ranged from mild to severe Recruited from five unrelated families	The study showed that 11/12 females were informative for the HUMARA l polymorphism, and in 10/11, the XCI pattern ranged from 50:50 to 65:35. The XCI pattern was not associated with clinical severity scores. The study demonstrated that clinical phenotype in Fabry disease heterozygous females was not influenced by XCI.
[56] Hübner et al. (2015)	9	A retrospective investigation of DNA methylation of the promoter region of the calcitonin receptor gene in Fabry patients	The sample consisted of the following: ➢ Six Fabry patients (non-ERT treated) ➢ Three Fabry patients (ERT treated) ➢ Six healthy controls	The study showed novel methylation at −78,504 CpG on both alleles in ERT-treated patients but not in non-ERT-treated and healthy controls. The calcitonin receptor methylation profile could indicate disease severity and potentially be an epigenetic biomarker in patients with ERT. However, further research needs to be performed in larger studies to verify this.
[58] Hossain et al. (2019)	36	To evaluate 36 heterozygous Fabry disease females using methylation studies of the GLA gene	The age range of the sample was 5 to 74 years old Clinical presentations varied	The study demonstrated an association with phenotype severity and lyso-Gb3 accumulation. Methylation-sensitive regions of the *GLA* gene were identified. Overall, there was a clear correlation between FASTEX scores, sphingolipid accumulation, and demethylation of the *GLA* gene.
[59] Echevarria et al. (2016)	56	To further understand the role of XCI in the clinical presentation in heterozygous females with Fabry disease	The study comprised 56 females aged (range) 45.75 years ± 15.15 (20–68 years) All patients had a confirmed diagnosis of Fabry disease	The study revealed that skewed XCI was identified in 29% of the samples. A correlation was found between XCI patterns in the blood and other samples. Skewing was responsible for differences in GAL levels, clinical severity scores, worsening of cardiomyopathy, and kidney disease. Global clinical severity change was assessed using the MSSI and DS3. Overall, the study suggested that XCI can impact the clinical phenotype in individuals with Fabry disease.
[60] Simoncini et al. (2016)	77	To investigate whether genetic polymorphisms in the mitochondrial genome could behave as disease modifiers in patients with Fabry disease	The sample consisted of 77 subjects (35 male and 42 female) Mean age: 42.23 ± 18.12 years The sample had five families (15 patients), and 62 cases were unrelated	The findings suggest that mitochondrial dysfunction and oxidative stress could be associated with Fabry disease. Haplotypes H and I and haplotype cluster HV were shown to be overexpressed in Fabry disease when compared to controls. Variation in the mitochondrial genome could contribute to the pathogenesis of Fabry disease. However, mitochondrial genetic variation and its role in the pathogenesis of Fabry disease warrant further investigation.
[61] Pan et al. (2016)	73	Evaluation of genotype–phenotype relationships in Fabry disease patients	Samples were taken from 73 patients from 53 unrelated families Control group consisted of 70 unrelated persons (30 women and 40 men) Clinical phenotypes were divided into classical and atypical Fabry disease	In the sample, 47 mutations were identified in all seven exons of the GLA gene. Twenty-three (23) novel mutations were reported. The findings from the study indicated that in female patients, the level of GAL activity could be associated with clinical severity. However, there was no association between GAL activity and genotype/phenotype in male patients. In male patients, genetic and non-genetic factors might be associated with the clinical phenotype.
[62] Di Risi et al. (2022)	5	Investigating methylation profiles in Fabry patients	Methylation status was profiled in 5 Fabry patients aged from 26 to 56 years All patients had the same *GLA* mutation (ex (exon 6 c.901C>G) on 6 c.901C>G)	The authors suggest that DNA methylome analysis might be beneficial for the management of Fabry disease. At the group level, the methylome profile in the Fabry disease group was not different from unaffected relatives. However, several genes were found to be differentially methylated in those with Fabry disease. This study showed that the methylation status of specific genes in Fabry disease may help to predict disease progression and organ involvement.
[63] Levstek et al. (2024)	99	Assessment of telomere length in patients with Fabry disease	Cross-sectional analysis; *n* = 99: ➢ Mean age ± SD: 47.3 ± 15.3 years ➢ Males (32.2%) Longitudinal analysis (5–10-year follow-up); *n* = 50: ➢ Mean age ± SD: 43.1 ± 15.1 years ➢ Males (32.0%) Patients were recruited from a Fabry disease clinic	The key findings from the study showed that leukocyte telomere length is not a disease indicator in older Fabry patients. The longitudinal analysis showed no association with disease progression and leukocyte telomere dynamics. There was also no difference in leukocyte telomere length in patients receiving disease-specific therapy and those not receiving therapy. The authors suggested that leukocyte telomere length may be more reflective of early-onset Fabry disease.
[64] Fu et al. (2022)	8	Examination of apabetalone treatment on inflammatory burden in cells isolated from Fabry patients treated with ERT	Patients were aged ≥18 years and receiving ERT	Inflammatory mediators are doubled in the plasma of Fabry patients treated with ERT. Apabetalone reduces inflammatory processes and oxidative stress in Fabry disease by modulating the transcription of inflammatory genes. Inflammatory and oxidative stress pathways may be important components in the pathogenesis of Fabry disease. The impact of apabetalone on modulating these pathways needs to be further verified in clinical trials.
[65] Wagenhäuser et al. (2022)	154	Association of XCI with clinical phenotype	Analysis was performed on 95 women (mean age [range]) who were an average of 53 years old (range 18–77) and 50 men (mean age [range]) who were an average of 42 years old (range 18–74) Cohort consisted of 104 women with Fabry disease and 50 men who were healthy or mutation carriers	The finding showed that neither clinical phenotype, GAL activity, nor lyso-Gb3 levels showed any difference in relation to XCI. In summary, the authors demonstrated that in women with Fabry disease, XCI patterns have limited use in understanding disease severity.
[66] Řeboun et al. (2022)	35	To assess the impact of XCI on the phenotype of Fabry disease patients by examining pitfalls in XCI testing	The sample of 35 female patients was from 22 families Average age (range): 56 years (24–72)	The study showed that using a combination of XCI methods alongside GAL activity assays improves the reliability of XCI testing. Methylation did not reflect the correct XCI status in one patient. A better understanding of tissue-specific and age-related XCI patterns can help to minimise potential bias when interpreting XCI studies in Fabry disease.

**Notes:** ^¥^ The clinical case presentation described a 49-year-old woman with suspected Fabry disease. ^Ψ^ Fibroblasts were derived from patients with Fabry disease. **Abbreviations:** BDP (bidirectional promoter); CpG (cytosine–guanine dinucleotide); DNA (deoxyribonucleic acid); DS3 (Fabry disease severity scoring system); ERT (enzyme replacement therapy); FASTEX (FAbry STabilisation indEX); GAL (α-galactosidase A); Gb3 (globotriaosylceramide); *GLA* (*alpha-galactosidase A*) gene; HUMARA (Human Androgen Receptor); MSSI (Mainz Severity Score Index); RNA (Ribonucleic Acid); SD (Standard Deviation); XCI (X—chromosome inactivation).

### 3.2. Thematic Analysis

A thematic analysis was performed on the data from the 20 identified articles. This enabled the generation of meaningful themes from the extracted data. Selecting keywords derived from the data led to the identification of eightcodes and their interconnections: (1) disease expression and modifying factors, (2) molecular mediators, (3) methylation patterns, (4) genetic pathways for management, (5) biochemical pathology, (6) epigenetic regulation, (7) XCI patterns and mechanisms, and (8) clinical severity (Appendix A). These codes led to the emergence of three main themes: (I) ‘*genetic modifiers’*, (II) ‘*methylation profiling*’, and (III) ‘*insights into X chromosome inactivation’*. Themes 1 and 2 had the highest frequency, emerging from seven studies. Theme (III) had the lowest frequency, emerging from six studies (Figure 2). Each of the themes is described in the next section.

Theme I: Genetic Modifiers

This theme emerged from seven studies that examined how methylation can influence disease severity. Fabry disease is thought to be associated with premature ageing [67]. One study investigated whether telomere length could have a role in predicting the clinical course of Fabry disease in 99 male and female individuals (mean age: 47.3 years) [63]. The study revealed no association between telomere length and disease stage. However, examination of telomere length may be more significant in early childhood and serve as a marker for monitoring disease progression during the initial stages of the disease [63]. In another study, the accumulative effects of bidirectional promoter (BDP) methylation levels alongside GLA mutations were associated with disease outcomes in Fabry individuals [49]. The BDP has regions susceptible to DNA methylation, which has a vital role in the expression of *GLA* loci. The authors demonstrated that DNA methylation of the BDP potentially caused by sphingolipids can influence disease manifestations. This study supports the premise that Fabry disease may not be solely due to *GLA* pathogenic variants, and other genetic modifications could play an important role. Inflammatory burden is prominent in Fabry disease, and one study demonstrated that the inhibition of epigenetic reader proteins can counteract this inflammation in individuals on enzyme replacement therapy (ERT) [64]. De novo somatic mosaicism was suggested to be a disease modifier in a 34-year-old male patient with late-onset Fabry disease with a classical GLA mutation and mild organ involvement [47]. Late-onset male patients with residual α-galactosidase activity present later, usually with cardiac and renal organ involvement [68].

When evaluating the genotype–phenotype relationship in Fabry disease, other evidence has shown that while α-GAL A activity may be associated with clinical phenotypes in female patients, the authors found no association between α-GAL A activity and genotype–phenotype relationships in male patients [61]. This study showed that in males, ocular activity was statistically associated with α-GAL A activity but not with genotype or clinical phenotypes. The study also demonstrated phenotypic variation in males from the same family and suggested that other disease modifiers (non-genetic or genetic) could influence the clinical phenotype in males with Fabry disease. Somatic mosaicism has been reported as a disease modifier in males with Fabry disease [69].

Examination of the mitochondrial genome (mtDNA) in 77 individuals with Fabry disease (35 male and 42 female) revealed that specific variants in mtDNA were more prevalent in individuals with Fabry disease when compared to neurotypical controls [60]. In particular, the study showed a disproportionate distribution of haplogroups H and I and cluster HV in individuals with Fabry disease. However, there was no association with gender, age of onset, or organ involvement. Nevertheless, the authors suggested that mtDNA haplogroups may have an important role in regulating oxidative stress in Fabry disease and warrant further investigation. Finally, a retrospective investigation of DNA methylation in the promoter region of the *calcitonin receptor* gene suggested that aberrant methylation of this gene could potentially serve as an epigenetic biomarker in individuals with Fabry disease on ERT [56].

Summary of Theme

From a clinical perspective, this theme highlights some important points. Reduction in telomere length has been associated with ageing [70]. A previous study has shown that telomere length is reduced in males with Fabry disease [67]. However, when viewed together with other studies, it appears that telomere length is not a reliable indicator of disease progression but may be more important during earlier disease stages [63]. Methylation of the BDP underscores the notion that Fabry disease may be influenced beyond pathogenic *GLA* variants. Mitochondrial DNA variants underscore the need for further research into the role of dysregulation in the mitochondrial epigenome in the pathogenesis of Fabry disease [60]. More recent evidence has shown that mitochondrial-related microRNAs (mitomiRs) are impaired in individuals with Fabry disease [71]. In biological samples derived from 63 individuals (mean age: 37 years, males: 45.6%), this study demonstrates that dysregulation in mitomiRs can affect oxidative phosphorylation, mitochondrial biogenesis, mitochondrial metabolism, and mitochondrial apoptosis, highlighting the essential role of mitomiRs as potential disease biomarkers in Fabry disease. When taken together, these findings suggest a critical role for mitochondrial dysregulation in the pathophysiology of Fabry disease [60,71]. In individuals on ERT, the inhibition of epigenetic reader proteins can reduce the inflammatory burden. Methylation at −78,504 CpG of the *calcitonin receptor* gene indicates its role as a modifier of pain suppression in Fabry disease [56]. However, the role of epigenetic reader proteins and −78,504 CpG methylation would also need to be validated in individuals pre-ERT to further substantiate their role as genetic modifiers of Fabry disease.

Theme II: Methylation Profiling

Methylation profiling emerged from seven studies describing evidence of gene methylation and its potential association with clinical severity in Fabry disease. Using DNA methylome analysis, the methylation status of approximately 850,000 CpG sites was investigated in five males with Fabry disease with identical *GLA* variants [62]. This study showed that certain genes were hypermethylated in males with Fabry disease, particularly the *zinc finger protein gene* (*ZFP57*), which plays a role in methylation at imprinting regions. This proof-of-concept study suggests that the episignatures of specific genes in Fabry disease could help better understand how methylation affects Fabry disease severity. In a study using endothelial cell lines derived from a 64-year-old male with Fabry disease, 15 signaling pathways were identified that could be more prone to methylation, alongside 21 genes with differential methylation profiles [52]. Decreased methylation was associated with the upregulation of two genes, namely, *collagen type IV alpha 1* (*COL4A1*) and *alpha 2* (*COL4A2*) genes [52]. Interestingly, this study showed that methionine levels were also increased in both human and animal tissues, which adds weight to the premise for a dysregulated methionine cycle in Fabry disease. Dysregulated methylation patterns could be crucial in predicting disease progression, particularly during early stages of diagnosis. Others have proposed a genetic algorithm that could facilitate early diagnosis, reaffirming the importance of early disease detection [50].

Methylation studies of the *GLA* gene can also provide insights into the disease. In 36 heterozygous females with Fabry disease, methylation-sensitive regions within the *GLA* gene were identified [58]. It would also be important to probe other areas of the *GLA* gene, especially when novel mutations emerge. One study identified a novel mutation (c.270C>G [p. Cys90Trp]) in the *GLA* gene [48] with predominant cardiac involvement. It would be helpful to examine the episignatures of this novel mutation in Fabry disease to determine whether methylation could predict the evolution of Fabry disease, particularly in cases of predominant cardiac involvement. This could be more important in males with Fabry disease who have a higher frequency of cardiovascular events, such as left ventricular hypertrophy, than females (53% versus 33%) [72]. Dysregulated methylation of the *GLA* gene was also shown to be associated with impairment of autophagy in Fabry disease [51]. The study showed that autophagy was abnormal in a female individual with severe disease, whereas the two sisters, who had mild symptoms, exhibited normal autophagic flux. Methylation profiling can reveal additional insights into disease trajectory, with methylation of the *GLA* gene associated with disease severity [46]. In this study, the affected female was found to have complete methylation of the non-mutated allele that contributed to early disease onset and a severe phenotype, in comparison to her sisters, whose alleles were non-methylated. When viewed all together, these studies show that methylation profiling could provide valuable insights into disease evolution and severity.

Summary of Theme

The evidence from the themes strongly supports a role for methylation profiling as a biomarker for disease severity. Methylation changes in genes such as *COL4A, GLA,* and *ZFP57* allow for the development of Fabry-specific episignatures. In other disorders, such as childhood-onset dystonia and Kabuki syndrome type 1, abnormal methylation profiles in the *lysine methyl transferase 2B* (*KMT2B*) gene can improve diagnostic accuracy [73]. In Fabry disease, abnormal DNA methylation patterns (episignatures) could help (I) to risk-stratify individuals depending on organ involvement, i.e., cardiac, renal, or (II) complement genetic testing, especially in heterozygous females or those with novel variants. However, further research in larger sample sizes would be required for a role of episignatures in improving diagnosis in Fabry disease. A recent evaluation of 16 episignatures in 10 neurodevelopmental disorders demonstrated that in the diagnostic setting, some episignatures may perform better than others [74]. The authors proposed that it would be prudent to examine regions that may escape from signatures [74]. This could have clinical relevance for Fabry disease because some genes may also escape XCI [30,75], leading to variability in episignatures. Theme I revealed that mitochondria are essential players in the pathophysiology of Fabry disease. In this theme, a study showed that abnormal methylation of the *GLA* gene can disrupt autophagy in Fabry disease [51]. Autophagy helps to remove dysfunctional mitochondria [76]. When viewed together, dysregulated autophagy and impaired mitochondrial homeostasis may synergistically contribute to disease severity. Dysregulated autophagy can lead to the poor clearance of dysfunctional mitochondria, and this may cause further cellular injury.

Theme III: Insights into X chromosome inactivation

Like most other lysosomal storage diseases, Fabry disease has incomplete genotype–phenotype correlations. Not only has clinical variability been noted for individuals within the same family, but it has also been noted for those from unrelated families with the same pathogenic variant [77]. Studies investigating XCI emerged from six studies. In one study, no correlation was found between XCI and clinical severity scores in females with Fabry disease [53], and the notion that XCI patterns cannot completely explain clinical severity in Fabry disease was also demonstrated by others [54,55]. However, one study from this review showed that XCI may explain the clinical variability in females with Fabry disease [59]. Others have suggested that XCI patterns have limited use in understanding disease severity in females with Fabry disease, especially when examining XCI in biological samples unaffected by disease [65]. This aspect was also considered in another study, which suggested that tissue-specific and age-related XCI patterns are important factors when interpreting studies of XCI in Fabry disease [66]. This work also recommends using a variety of XCI assays to help minimise potential bias in the interpretation of XCI studies in Fabry disease.

Summary of Theme

The findings from this theme illustrate the complexity of unravelling the clinical variability in Fabry disease. The results from XCI studies are inconsistent, and sampling bias in assays used to measure XCI patterns in blood or buccal tissue may be a contributing factor to this inconsistency. Patterns in blood or buccal tissue may not necessarily reflect the XCI patterns in organs affected in Fabry disease. To circumvent this issue, transcriptome and exome sequencing data from blood offer another method to accurately measure XCI status and skewing [78]. Although new methods may help further our understanding of the complex interplay between XCI and clinical variability in Fabry disease, they still may not offer a complete explanation. While XCI may play a role in explaining some of the variability, it is likely that other epigenetic factors also significantly influence disease manifestations in individuals.

## 4. Discussion

According to the literature, this is the first narrative review and thematic analysis of studies examining methylation in Fabry disease. Three themes emerged that can help further our understanding of the clinical manifestations of Fabry disease. The thematic analysis showed that (I) telomere length, especially in early disease stages, (II) BDP methylation by sphingolipids, (III) epigenetic reader proteins, (III) mtDNA haplogroups H and I and cluster HV, and (IV) DNA methylation of the promoter region of the *calcitonin receptor* gene in individuals on ERT could potentially function as molecular mediators of clinical variability in Fabry disease and is summarised in Figure 3. Although these molecular mediators play a role in influencing the clinical phenotype in Fabry disease, the findings are associative in nature and should be interpreted with caution. Given the small and heterogeneous cohorts, the lack of replication of studies, and variability in methods, the translational evidence of the findings should be tempered, and we are unable to confirm whether one molecular mediator is more important than the other in driving clinical variability. Further research in multicentre, longitudinal studies with larger sample sizes would be needed to confirm whether molecular mediators, such as genetic modifiers and/or DNA methylation, could be used as reliable clinical epigenetic biomarkers in individuals with Fabry disease. The findings from this review can be broadly categorised into three levels: (I) findings with relatively solid support, (II) emerging hypotheses, and (III) gaps in the literature. These are discussed in the next section.

### 4.1. Genetic Modifiers

When viewed broadly across the epigenetic landscape of Fabry disease severity, genetic modifiers could play a role as disease biomarkers.

#### 4.1.1. Findings with Relatively Solid Support

Evidence shows that specific biomarkers and clinical tools can be used to assist in the diagnosis and stratification of Fabry disease in males. For example, the Fabry Outcome Survey (FOS) [34] and Fabry Registry [79] disease registries could help risk-stratify individuals with Fabry disease and improve prognosis. Levels of α-GAL A are essential in determining the pathogenicity of a variant in males, and levels of plasma lyso-Gb3 are a helpful diagnostic biomarker in males [35]. Interestingly, our analysis also revealed a potentially higher inflammatory burden in individuals with Fabry disease [64]. Pro-inflammatory cytokines have been implicated in the pathophysiology of Fabry disease, reinforcing the role of inflammation in this condition [80,81,82]. Evidence has shown that peripheral blood mononuclear cells in individuals with Fabry disease have increased levels of pro-inflammatory cytokines [83]. In Fabry disease, glycolipid deposits in lysozymes trigger chronic inflammation, leading to progressive organ damage [82].

#### 4.1.2. Emerging Hypothesis

The review showed a promising yet preliminary hypothesis for understanding how genetic modifiers influence Fabry disease. While plasma lyso-Gb3 is helpful in diagnosing hemizygous males, its ability to detect disease burden in heterozygous females and those with non-classical GLA variants [35] is questionable. Normal levels of plasma lyso-Gb3 could be considered in later-onset forms [17]; however, plasma lyso-Gb3 might not be able to accurately distinguish between early- and later-onset forms of Fabry disease in a newborn screening study [84]. In non-classic and female individuals, lyso-Gb3 should be viewed together with other diagnostic and monitoring tools [85].

Methylation at site 78,504 CpG in the promoter region of the *calcitonin receptor* gene could potentially serve as an epigenetic biomarker for individuals with more severe Fabry disease [56]. However, the small sample size in this study precludes the drawing of meaningful inferences. Methylated lyso-Gb3 isoforms could help identify individuals with later-onset variants [86]. However, greater emphasis needs to be placed on longitudinal, multicentre studies before DNA methylation of the promoter region of the *calcitonin receptor* gene and/or methylated lyso-Gb3 isoforms can be considered clinically relevant.

While there are biomarkers for Fabry disease [87], the biochemical biomarkers for detecting Fabry disease in heterozygous females are limited [84,85]. In this context, genetic modifiers may be more relevant in heterozygous females, who present later than males, with variable disease severity [88]. Genetic modifiers could, therefore, have an increasingly important role in modifying the clinical presentation in females with the same *GLA* pathogenic variant. Preliminary evidence from a proteomics-based analysis has revealed a role for interleukin 7 (IL-7) in stratifying individuals, especially those with the non-classical form [80]. However, further studies in a larger cohort are needed to determine whether a proteomic signature can be clinically utilised as a diagnostic tool in Fabry disease. Profiling mitochondrial dysregulation through mitochondrial DNA haplogroups in 77 individuals with Fabry disease provides preliminary evidence for mitochondrial genetic variation in disease variability [60]. The study of mitomiRs further opens new avenues for profiling metabolic processes in Fabry disease [69] but may also help to identify metabolic mechanisms underpinning other lysosomal storage disorders [89].

#### 4.1.3. Gaps in the Literature

The present review showed that studies on reliable biochemical biomarkers for detecting and monitoring disease progression in heterozygous females are still lacking [84,85]. Furthermore, the value of plasma lyso-GB3 in (I) diagnosing non-classical and female patients and (II) differentiating between disease severity or onset remains unclear. The literature also indicates that while epigenetic modifiers and inflammatory markers show promise, studies validating epigenetic and inflammatory markers, such as CpG methylation IL-7, for routine clinical testing are lacking. A targeted research approach is required to address these gaps, especially for heterozygous individuals.

### 4.2. Differential Methylation Profiles

The thematic analysis also revealed that specific genes exhibit differential methylated profiles in males with Fabry disease [52,62].

#### 4.2.1. Findings with Relatively Solid Support

In Fabry disease, evidence from the FOS has shown that the incidence of cerebrovascular events ranges from 11.1 to 25% in males and from 15.7 to 21% in females [34,90,91], with these events more frequently occurring at a younger age in males [91]. The vasculopathy in Fabry disease is due to the deleterious impact of lyso-Gb3 on blood vessels and its interaction with the nitric oxide pathway [92,93]. A recent single-centre study has evaluated the cardiac phenotypes in 42 individuals with Fabry disease [94]. This study showed that even at early disease stages, cardiac abnormalities are present, and those with classical variants had a statistically elevated risk of experiencing an adverse cardiovascular event. The authors suggest that genotypes modulate functional cardiac remodelling in Fabry disease [94].

#### 4.2.2. Emerging Hypothesis

In the current study, preliminary evidence suggests that decreased methylation of *COL4A1* and *COL4A2* may play a role in disrupting blood vessel architecture and contribute to vasculopathy in individuals with Fabry disease [52]. These genes are responsible for maintaining basement membranes and are involved in a range of multi-organ disorders, including cerebrovascular pathologies [95]. A broad spectrum of cerebrovascular manifestations has been identified in those with *COL4A1* and *COL4A2* pathogenic variants [96]. Genome-wide methylation profiling in five male individuals with Fabry disease also revealed differential methylation profiles [62]. This proof-of-concept study suggests that episignatures could be used as diagnostic markers. While these results are promising and suggest that specific genes are differentially methylated in Fabry disease, the findings should be interpreted with caution. One study was conducted using endothelial cell models derived from a male individual with Fabry disease and does not entirely capture the complexity of the disease [52]. Moreover, the small sample size in studies limits their ability to identify specific episignatures that can reliably discriminate between individuals with Fabry disease [62].

Emerging evidence also suggests that a dysregulated methionine cycle may also contribute to differential methylation profiles in individuals with Fabry disease [52], and this mirrors the findings in other X-linked disorders [25,97]. When viewed together, differential methylation of *COL4A1* and *COL4A2* genes suggests an association with cerebrovascular vulnerability in males with Fabry disease. This, coupled with a dysregulation in the methionine cycle, allows further insight into how methylation can affect disease manifestations in Fabry disease and draws parallels with other X-linked disorders.

#### 4.2.3. Gaps in the Literature

Further research is needed to better understand the epigenetic landscape of Fabry disease. Most notably, evidence suggesting a causal relationship between altered methylation of *COL4A1* and *COL4A2* genes and cerebrovascular pathologies in Fabry disease remains to be firmly established. Unlike some other X-linked disorders, the role of a dysregulated methionine cycle in Fabry disease is still underexplored. The lack of data also extends to gender-specific epigenetic mechanisms, especially the literature on the epigenetic mechanisms in heterozygous females, where disease patterns are more variable. While differential methylation profiles in Fabry disease are a promising area of research, longitudinal studies are needed to assess whether modifying methylation profiles could help to improve clinical outcomes in Fabry disease.

### 4.3. X Chromosome Inactivation

#### 4.3.1. Findings with Relatively Solid Support

In Fabry disease, evidence from studies on XCI in females with Fabry disease varied. There were inconsistent results regarding the influence of XCI on clinical manifestations [53,54,55,59]. The heterogeneous clinical picture in females often leads to misdiagnosis. Indeed, about 25% of individuals in the FOS were misdiagnosed [34]. Misdiagnosis can also lead to delays in initiating disease-modifying therapies, which is a crucial aspect because evidence has shown that timely intervention using ERT can lower the risk to organs, regardless of the disease type [98]. In heterozygous females with Fabry disease, XCI may play a role in clinical variability.

#### 4.3.2. Emerging Hypothesis

Variability in XCI patterns is influenced by age-related changes in XCI and tissue-specific organ manifestations [66]. Methodological differences in the assessment of XCI could influence the outcome of studies [75]. Approaches such as ultra-deep DNA methylation analyses have been proposed as promising tools to facilitate diagnosis in females with Fabry disease [33,75], and this could theoretically optimise the time for treatment initiation with disease-modifying treatments, such as ERT and chaperone therapies [99], before irreversible organ damage occurs. Other factors implicated in clinical variability may be attributed to the number of genes that escape XCI [30,75], which lead to differences at the cellular level even within the same tissue [16]. This phenomenon adds another layer to phenotypic variability in Fabry disease. However, this hypothesis remains preliminary and would require further investigation before the clinical significance of genes escaping XCI can be established.

#### 4.3.3. Gaps in the Literature

Despite insights into XCI, gaps remain in understanding the precise role of XCI in the clinical variability in Fabry disease. The purported mechanisms of (I) age-related changes, (II) tissue-specific mechanisms, and (III) XCI escape have not been fully elucidated. Inconsistencies across studies also underscore the need for research into standardised methods to assess XCI patterns accurately, and while ultra-deep methylation shows promise, this technique has not been validated in individuals with Fabry disease.

### 4.4. Limitations

The lead author (J.S.) performed the narrative review; however, the review was not undertaken blindly, as preliminary discussions with the third author (U.R.) addressed key issues and concepts. These included unmet needs, especially relating to genetic modifiers, in female individuals with Fabry disease. The SANRA scale was used as a self-assessment tool to evaluate the quality of the review. Higher scores indicate better quality, with a maximum of 12. This review was prepared in accordance with the SANRA framework, and the self-assessment score is presented in Appendix A. Another limitation is the interpretative nature of thematic analysis. Colour coding of words/terms that led to the emergence of themes was performed manually by the lead author. This process partly followed the method of conducting a thematic analysis used in a previous study by the first author, which examined autonomic dysregulation in another X-linked disorder, RTT [100]. However, in this study, the thematic analysis was performed by a single author (J.S.), increasing the risk of bias that could have unintentionally influenced the identification of themes. This limitation in the thematic analysis can affect the generalisability of the findings. However, to reduce the bias, the data extraction and thematic framework were reviewed by another author (U.R.), and any disagreements were resolved before data extraction, and the thematic framework was finalised. The following strategies were also employed to minimise interpretive bias: (I) adopting a structured framework for the thematic analysis as described [43], (II) using 6Rs to avoid the drift of keywords and terms, (III) employing the principles of 4Rs to assist in the identification of themes, (IV) using inductive coding and a data-driven approach to allow themes to emerge naturally, and (V) having the thematic framework reviewed by another author (U.R.) to ensure consistency of keywords and codes.

In summary, while the use of the SANRA assessment and structured framework for thematic analysis strengthened the methodological rigour of the review, the absence of independent data extraction can introduce bias. However, the third author (U.R.) reviewed the data extraction and thematic framework. Disagreements were resolved before finalising the thematic framework. Inter-coder statistics were not performed. However, as mentioned previously, this does not necessarily undermine the validity of the coding [101]. The limitations of the underlying literature were also clearly acknowledged, including small and heterogeneous cohorts, lack of replication, and variability in methods. Despite these constraints, the review collected and organised a fragmented and limited body of evidence on an emerging topic in Fabry disease.

### 4.5. Future Directions

Artificial intelligence (AI) algorithms have been adopted to address medical issues in individuals with Fabry disease [102]. These range from supporting diagnosis using electronic health records to a better understanding of organ-specific disease progression [102]. Promoter-level transcriptome analysis alongside artificial intelligence (AI) algorithms was employed to identify the *chimerin 1* (*CHN1*) gene as a potential biomarker for cardiac issues in 15 male individuals with Fabry disease [103]. However, this finding would need to be replicated in a larger sample to assess whether this biomarker could be used in clinical settings. In this view, future studies of epigenetics could utilise AI methodologies to enhance our understanding of Fabry disease.

## 5. Conclusions

The combined narrative review and thematic analysis revealed important insights into the epigenetic mechanisms of Fabry disease. This mapping exercise may help to inform health professionals and researchers working in Fabry disease. Robust longitudinal and multicentre studies would be essential before any proposed epigenetic biomarker can be considered clinically reliable and adopted into standard care. Ultimately, the goal of unravelling epigenetic mechanisms is to facilitate the diagnosis and management of Fabry disease, especially at early stages before symptoms become entrenched and irreversible organ damage occurs.

## Figures and Tables

**Figure 1 cimb-47-00855-f001:**
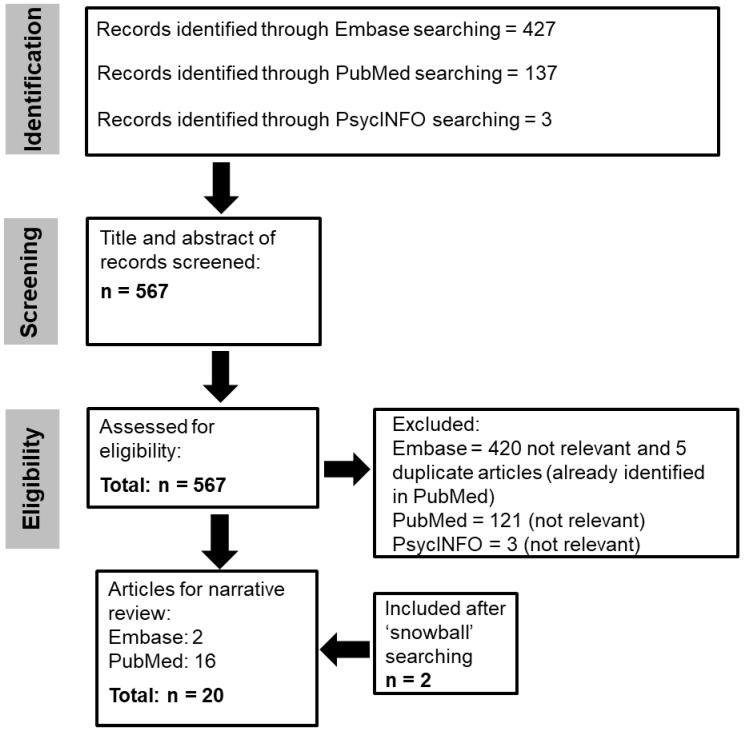
Flowchart of the search.

**Figure 2 cimb-47-00855-f002:**
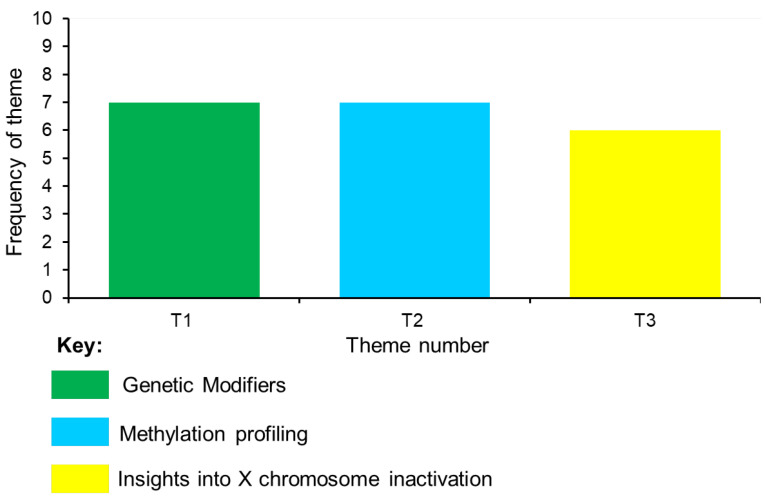
Frequency of themes.

**Figure 3 cimb-47-00855-f003:**
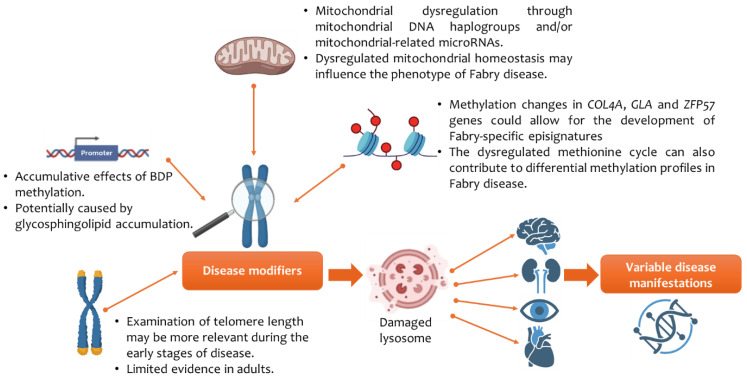
Epigenetics and its impact on clinical variability in Fabry disease. Abbreviations: alpha-galactosidase A (GLA); bidirectional promoter (BDP); collagen type IV alpha gene (*COL4A*); zinc finger protein gene (ZFP57). Figure created using images from BioRender (https://biorender.com/).

## Data Availability

The data in this narrative review was derived from articles openly available in the Embase, PsycINFO (https://www.ovid.com/), and PubMed (https://pubmed.ncbi.nlm.nih.gov/) databases.

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
