# Peer review of "Epigenetic Mechanisms in Fabry Disease: A Thematic Analysis Linking Differential Methylation Profiles and Genetic Modifiers to Disease Phenotype"

_cimb, 2025, doi:10.3390/cimb47100855_

Round 1
Reviewer 1 Report
Comments and Suggestions for Authors
The manuscript "Epigenetic Mechanisms in Fabry Disease: A Thematic Analysis Linking Differential Methylation Profiles and Genetic Modifiers to Disease Phenotype" by Singh, Santosh, and Ramaswami is a review that collects and organizes a scattered and limited body of evidence on an emerging topic in Fabry disease. The attempt to apply structured frameworks (SANRA, thematic analysis) is commendable, and the acknowledgment of methodological limitations is appreciated.
However, much of the discussion simply restates findings from the included studies without deeper critical analysis. The authors should consider distinguishing more clearly between stronger evidence (replicated or based on larger cohorts) and preliminary or single-study findings.
The current wording suggests that several proposed biomarkers are promising, whereas most evidence is exploratory and based on small sample sizes. The conclusions should therefore be substantially moderated.
The section on artificial intelligence is not clearly linked to the review results and appears speculative. It would be more appropriate to shorten it or move it into a brief forward-looking note.
Although some methodological issues are acknowledged, the limitations section currently reads as self-justifying. It should be stated more explicitly that data extraction and coding conducted by a single author, without independent verification, may introduce bias. The supplementary tables provide a clear overview of the SANRA assessment, but this does not mitigate the concern about single-author bias. In addition, the limitations of the underlying literature should be more clearly recognized, including small and heterogeneous cohorts, lack of replication, and variability in methods.
To improve clarity, the discussion could be structured into three levels:
a) Findings with relatively solid support.
b) Preliminary hypotheses.
c) Clear gaps in the literature.
Such structuring would increase the usefulness of the review for researchers planning future studies.
Finally, the conclusions are currently too ambitious and speculative relative to the available evidence. They should be toned down, with an emphasis that this represents only a first mapping exercise. Greater emphasis should also be placed on the need for longitudinal, multicenter studies before any biomarker or epigenetic mechanism can be considered clinically relevant.
Author Response
Please see attached file with responses to Reviewer 1

Reviewer 2 Report
Comments and Suggestions for Authors
In their Ms ID cimb-3900561 – “Epigenetic Mechanisms in Fabry Disease: A Thematic Analysis Linking Differential Methylation Profiles and Genetic Modifiers to Disease Phenotype”- the Authors examined DNA methylation and genetic modifiers as key factors modifying clinical variability in Fabry disease (FD), an X-linked lysosomal storage disorder characterised by impaired metabolism of glycosphingolipids whose accumulation causes irreversible organ damage and life-threatening complications. More specifically, from the present review, the following variables appear to be linked to FD phenotype variability: i) telomere length especially in early disease stages, (ii) BDP methyla-tion by sphingolipids (iii) epigenetic reader proteins, (iv) mtDNA haplogroups H, I and cluster HV and (v) DNA methylation of the promoter region of the calcitonin receptor gene in individuals on ERT. Overall, the review is well written and structured, and the points of view are quite reasonable and straightforward. Nonetheless, a few issues should be addressed, specifically:
- Abstract “Conclusion” for a narrative review is quite an overstatement (“This thematic review demonstrates that DNA methylation and genetic modifiers are key factors modifying clinical variability in Fabry disease”). Hence, I would strongly suggest replacing the verb “demonstrates” into “shows” or equivalent ones. This also applies to corresponding sections in the Discussion.
- Although Table 1 is informative, it is quite difficult to read. I would suggest either paying more attention to the synthesis of the “relevant findings”column or to modify the table format.
Author Response
Please see attached file with responses to Reviewer 2

Reviewer 3 Report
Comments and Suggestions for Authors
The manuscript “Epigenetic Mechanisms in Fabry Disease…” by Jatinder Singh et al. is aimed to review the role of DNA methylation and epigenetics on the clinical phenotype in Fabry disease. Authors summarized the data on three themes: genetic modifiers, methylation profiling, and X-chromosome inactivation. They noted that DNA methylation and genetic modifiers are key factors modifying clinical variability in Fabry disease.
The review involves a relevant issue and seems very useful. The article is well written and allows understanding the problem. The methods used in the work are adequate. A sufficient number of relevant articles have been cited, however, references must be edited according to the journal requirements.
To improve the quality of the manuscript, the authors should address the following issue. The table 1 should be edited, namely, 'Assessment Methods' and 'Relevant Findings' columns should be enlarged (widened).
After minor revision, the article could be accepted for publication.
Author Response
Please see attached file with responses to Reviewer 3

Round 2
Reviewer 1 Report
Comments and Suggestions for Authors
Thank you for carefully revising the manuscript according to the suggestions. The paper is now improved, and I have no further comments to add. In my opinion, it is ready for publication.